# Epidemiology of alcohol use and alcohol use disorder among female sex workers in Mbeya City, Tanzania

**Andrew Kapaya Augustine**[1]*, **Lucas Maganga**[2], **Joel Msafiri Francis**[3]

**1** Department of Epidemiology and Biostatistics, School of Public Health and Social Sciences, Muhimbili University of Health and Allied Sciences, Dar es Salaam, Tanzania, **2** Mbeya Medical Research Centre, National Institute for Medical Research, Mbeya, Tanzania, **3** Department of Family Medicine and Primary Care, School of Clinical Medicine, Faculty of Health Sciences, University of the Witwatersrand, Johannesburg, South Africa

* andyjmore@gmail.com

**Data Availability Statement:** Data relevant to this paper are available from the Harvard Dataverse at https://doi.org/10.7910/DVN/ZSYQ7T.

## Abstract

Alcohol misuse is a global concern, contributing to 5.3% of total deaths and 132.6 million disability-adjusted life years worldwide. In Sub-Saharan African countries, the prevalence of Alcohol Use Disorder (AUD) has risen, especially among female sex workers, due to increased availability and advertising. However, there are limited studies on alcohol use and AUD among female sex workers in Tanzania. This study aimed to determine the prevalence, patterns, and factors associated with alcohol use and AUD among sex workers in Mbeya city, Tanzania. In this cross-sectional study, 212 female sex workers in Mbeya city, Tanzania, seeking enrolment in the National Institute for Medical Research Mbeya Medical Research Centre's registration cohort from July to November 2022. Structured interviews covered socio-demographics, alcohol screening (AUDIT-C and Timeline Follow Back Calendar), and sexual behaviours data. Data were analysed using Stata version 17. Descriptive analysis assessed alcohol consumption and AUD prevalence. Factors associated with alcohol use and AUD at bivariate analysis were identified using Chi-square/Fisher's exact tests. All variables with p-value ≤ 0.20 were entered into a multivariable logistic regression model to identify factors associated with alcohol use and AUD. Among 212 participants, 86.6% reported alcohol use in the past 12 months, 85% in the past 30 days, and 98.5% met AUD criteria. Factors linked to recent alcohol consumption included primary education or higher, income above the median, and more than 10 sexual partners. Education level, marital status, income, and having dependents were significantly associated with heavy drinking episodes. The prevalence of AUD, alcohol use, and heavy episodic drinking were high among female sex workers in Mbeya city. Socio-demographic factors and risky sexual behaviours were associated with alcohol use and heavy episodic drinking highlighting the need for targeted interventions to combat alcohol abuse among female sex workers within the HIV program.

**Funding:** The authors received no specific funding for this work.

**Competing interests:** The authors have declared that no competing interests exist.

## Introduction

Alcohol abuse has emerged as a significant global health concern, contributing to a staggering number of deaths and disability-adjusted life years (DALYs) worldwide. Statistics reveal that alcohol-related causes account for nearly 3 million deaths annually, representing approximately 5.3%% of all recorded deaths. Surprisingly, this places alcohol as a leading cause of mortality, surpassing other prominent diseases such as HIV, tuberculosis, and diabetes [1, 2].

Alcohol use disorder (AUD) presents a significant and complex challenge in low-income and middle-income countries, giving rise to a range of interconnected issues such as the spread of diseases, injuries, domestic violence, and the perpetuation of poverty [3, 4]. Extensive research has shed light on the correlation between alcohol consumption and the risk of HIV infection, unveiling a concerning link that underscores the importance of addressing this matter [3]. Additionally, investigations have revealed a distressing pattern, indicating that alcohol use contributes to a higher incidence of gender-based violence among female sex workers, highlighting the urgent need for interventions and support systems [5–8]. It is crucial to comprehensively address these intertwined concerns in order to promote healthier communities and improved well-being.

Alcohol use disorder (AUD) presents a significant concern within specific demographic groups, notably among young individuals, including students [9], and sex workers [3, 10]. Furthermore, individuals living in impoverished conditions and those who engage in alcohol consumption exhibit heightened vulnerability to risky behaviours [11]. It is crucial to acknowledge that alcohol consumption in these susceptible populations may exacerbate existing health-related challenges [8, 12].

Over the past years, Sub-Saharan African nations have witnessed a discernible increase in AUD, primarily attributed to amplified accessibility and pervasive marketing campaigns associating alcohol intake with success [1, 3]. This trend particularly impacts young adults, including university students [3, 9], and both female and male sex workers in urban centres such as Dar es Salaam, where the estimated population of sex workers reaches approximately 150,000 [13]. Moreover, bar workers also face elevated risks due to the prevalent alcohol culture in these environments [14, 15].

Despite some research exploring alcohol use among high-risk populations in Tanzania, such as bar and hotel workers, the prevalence and patterns of alcohol use and alcohol use disorder among female sex workers in Mbeya, Tanzania largely unexplored. The study findings will play a pivotal role in shaping targeted strategies aimed at reducing alcohol consumption and preventing AUD, tailored specifically to address the unique needs of female sex workers in Mbeya.

## Methods

### Study design

We conducted a cross-sectional study among female sex workers seeking enrolment in the ongoing cohort study run by the National Institute for Medical Research, Mbeya Centre, in Mbeya, Tanzania.

### Study setting

The research was conducted in the Mbeya region, situated in the south-western of Tanzania's Southern Highlands. With an approximate population of 2.71 million individuals [16], the study specifically focused on the Mbeya City Council, one of the seven district councils in the

region. The city council has a population of around 385,279, with females constituting 53% of the population, according to the 2012 census data [16].

Significantly, the Mbeya region carries a considerable burden of HIV, with an estimated prevalence of 9.2% [17]. Moreover, the region serves as a crucial transportation corridor to Southern African countries, and the city of Mbeya is estimated to be home to 10,152 female sex workers and the similar report indicated that they had a HIV prevalence of 26% compared to 11% in 2013 [18].

The Mbeya City Council offers HIV services according to the National AIDS Control Program (NACP) under the Ministry of Health, Community Development, Gender, Elders, and Children. The services aim to achieve the sustainable goal of controlling the HIV epidemic and achieving zero transmission by 2030. These services are provided through various healthcare facilities., including dispensaries and regional referral hospitals, in collaboration with community-based HIV services, thereby ensuring comprehensive care [19]. The NIMR-Mbeya Medical Research Center plays a significant role in recruiting high-risk women for a registration cohort. This cohort provides participants with various interventions and services while offering opportunities to engage in studies that inform HIV programs and services.

## Study population

The study population consisted of female sex workers who were newly mobilized for screening and enrolment in the HIV high-risk women cohort at the NIMR-Mbeya Medical Research Center, located at the Mbeya Zonal Referral Hospital in Mbeya City. The inclusion of this specific population was driven by their elevated risk status and their potential to contribute valuable insights to inform HIV programs and services.

By focusing on this targeted population, the study aimed to acquire essential knowledge regarding the prevalence, patterns, and determinants associated with alcohol use and alcohol use disorder (AUD) among sex workers in Mbeya, Tanzania. Ultimately, this research aspires to shape targeted strategies that effectively reduce alcohol consumption, prevent AUD, and cater to the distinct needs of this vulnerable population.

## Sample size estimation

To ensure an adequate representation of the population of interest, the sample size for this study was calculated using the single standard proportion formula. This formula, denoted by $N = Z^2 P (1-P)/E^2$ where Z standard normal deviation corresponding to 5% = 1.96, prevalence for hazardous drinking was 14.3% and Marginal error of the study = 5% and adjusted for 10% non-response [20]. A total of 208 Female Sex Workers were expected to be interviewed but 212 turned for the interviewed and were included in this study.

## Mobilization of the NIMR-Mbeya female sex worker registration cohort

In collaboration with community leaders, members of the community advisory board, and the recruitment team, the NIMR-Mbeya Research Centre conducted a thorough mapping of potential recruitment sites. These sites included locations characterized by high-risk HIV behaviour patterns, such as hotspots for transactional sex activities, areas with a high prevalence of transmission, restaurants, guest houses, bars, and local breweries. Individuals identified as being at a heightened risk were invited to join the registration cohort established by the centre. This registration cohort enables participation in future HIV studies, contributing to the development of effective programs and services. It is essential to acknowledge that the research activities received approval from the national ethics committee.

## Sampling of the study participants

Female sex workers who presented themselves at the NIMR-Mbeya Research Centre for potential enrolment in the registration cohort study were consecutively recruited to participate in this study from 12th July 2022 to 06th December 2022. This study was conducted independently and holds no direct affiliation with the registration cohort study.

## Eligibility criteria

The study involved the inclusion of participants based on the following criteria:

i.  Females aged 18 years and above.

ii.  Engaged in commercial sex within the past 12 months.

iii.  Willingness to provide informed consent for study participation.

Exclusion criteria encompassed individuals who had undergone screening and enrolment in the registration cohort study.

## Study variables

**Outcome (dependent) variables.** In this study, we considered the following outcome variables:
Alcohol Use:

a.  Alcohol use in the last 12 months.

b.  Amount of alcohol consumed in the last 1 month, evaluated using the Timeline Follow-Back (TLFB) method.

Alcohol Use Disorder (AUD) in this study it was defined by the AUDIT-C score $\geq 4$.
**Exposure (independent) variables.** The exposure variables investigated in this study encompassed the socio-demographic characteristics and sexual behaviours of Female Sex Workers.

## Data collection tools

For data collection, participants who agreed to take part in the study completed a structured questionnaire. The questionnaire consisted of various components, including socio-demographic information, alcohol screening tools (AUDIT-C and the TLFB calendar), and an assessment of sexual behaviours. Below, we provide specific details about the AUDIT-C and TLFB calendars.

**AUDIT-C.** The AUDIT-C, known as the Alcohol Use Disorders Identification Test, is a well-recognized and validated screening instrument specifically designed to assess alcohol consumption patterns. It effectively categorizes individuals into different groups based on their drinking behaviour, including low-risk drinkers, high-risk drinkers, and those with active alcohol use disorders. This instrument plays a crucial role in identifying and evaluating alcohol-related issues, providing valuable insights for effective intervention and treatment strategies. Classification is based on the AUDIT-C scores, which range from 0 to 12. A score of 0–3 indicates low-risk drinking (no AUD), while a score between 4 and 12 suggests high-risk drinking (AUD) [21].

**TLFB calendar.** The TLFB calendar is a validated method employed for assessing drinking patterns among clinical and nonclinical populations [21]. During the study, Female Sex Workers retrospectively estimated their drinking behaviour for the month preceding the

interview using the TLFB calendar. The interviewer then recorded industrial brew reported by Female sex Workers and converted the reported number of drinks consumed into standard units, utilizing the List of Alcohol and their Standard Drinks. The concentration of alcohol in local beverages used in this study were profiled in Tanzania [22]. By examining the frequency and quantity of alcohol consumed, we gained valuable insights into the patterns, variability, and magnitude of individuals' drinking behaviours.

## Validation of data collection tools

The data collection tools used to assess alcohol use and alcohol disorders, including the AUDIT-C and TLFB calendar, have been validated for use in the specific Tanzanian context [21].

## Reliability

Pilot tests were conducted with clients who were already enrolled in the registration cohort at the NIMR Mbeya Medical Research Centre to ensure the reliability of the data collection instruments, including the questionnaire, AUDIT-C, and TLFB calendar. These pilot participants were selected based on their similarities to potential future cohort recruits. The pilot tests aimed to assess the clarity and comprehensibility of the questions, identify any potential issues, unforeseen interpretations, and cultural objections, and make necessary adjustments to the content and wording of the data abstraction form.

## Data collection procedure

The study utilized an interview-based questionnaire to gather information from the study participants, who were Female Sex Workers (FSWs). Before enrolling the FSWs into the study, the principal investigator and/or the research assistant provided a detailed explanation of the alcohol study to each FSW visiting the center. Following the briefing, the FSWs were invited to provide written consent by completing a consent form. Each participant signed consent form for their records. The data collection process aimed to achieve the desired sample size within a timeframe of four months.

## Data management and analysis

Regarding data management, the completed questionnaires underwent a thorough review to ensure accuracy and consistency. The information from the questionnaires was entered twice into Epi-data 3.1 software to minimize data entry errors. The two datasets were then compared to identify any discrepancies, which were resolved by carefully reviewing the questionnaires. Once the data were cleaned and validated, they were imported into Stata version 17 for analysis. Prior to the analysis, the data were coded in the following manner: Alcohol consumption within the past year was treated as a binary variable without any further coding. The AUDIT-C scores were classified as 0–3 (indicating no Alcohol Use Disorder—low-risk drinking) or $\geq 4$ (indicating Alcohol Use Disorder—risky drinking). The Timeline Follow-Back (TLFB) data, which recorded the amount of alcohol consumed, were categorized as either 6 drinks (indicating no binge drinking) or $>6$ drinks (indicating binge drinking).

For the data analysis, all statistical analyses were performed using Stata version 17.

## Ethics statement

Formal written Informed consent was obtained from each participant. The study questionnaires were securely stored in a restricted location, accessible only to the principal investigator

and research assistants involved in the study and softcopies were secured using password accessed only by authorized personals.

In the event that participants were identified as having AUD were referred to the psychologist at the Mbeya Zonal Referral Hospital for further evaluation and treatment.

The study received ethical approval from Muhimbili University of Health and Allied Sciences (Ref. No.DA.282/298/01.C/) and the Mbeya Medical Research and Ethics Committee (Ref No:SZEC-2439/R.A/V.1/143a).

## Results

### Socio-demographic characteristics

A comprehensive set of interviews was conducted, involving a total of 212 individuals who sought enrolment in the registration cohort at MMRH. Among the diverse participant pool, a noteworthy 94.8% resided in Mbeya City Council, with 112 individuals (54.8%) falling into the age bracket of 25 or older. In terms of educational attainment, it was revealed that 72 individuals (34%) had successfully completed their primary education, while a further 84 individuals (39.6%) had surpassed this milestone and attained a secondary school education or higher. 148 individuals (69.8%) reported being single. Considering the financial aspect, it was discovered that 110 individuals (51.9%) earned a median income of $90,000 or less. Additionally, a significant portion of the participants, comprising 130 individuals (61.3%), carried the responsibility of dependents. For a more comprehensive breakdown of this valuable information, please refer to **Table 1**.

### Prevalence of alcohol use

Within the population of female sex workers, a comprehensive analysis revealed striking levels of alcohol consumption, as demonstrated by the insights presented in **Table 2**.

**Table 1. Socio-demographic profile of female sex workers in Mbeya City Jul- Nov 2022.**

| Characteristic | Categories N = 212 | n | % |
|---|---|---|---|
| **Location** | Mbeya City | 201 | 94.8 |
| | Others | 11 | 5.2 |
| **Age (Years)** | 18–24 years | 100 | 47.2 |
| | 25 and above | 112 | 52.8 |
| **Education** | Incomplete primary school and never | 56 | 26.4 |
| | Primary school | 72 | 34.0 |
| | Secondary school and above | 84 | 39.6 |
| **Relationship status** | Single | 148 | 69.8 |
| | In a relationship (including married and cohabiting) | 64 | 30.2 |
| **Income (Tshs)** | Median (90000 and below) | 110 | 51.9 |
| | Above median income (>90000) | 102 | 48.1 |
| **Dependents** | Yes | 130 | 61.3 |
| | No | 82 | 38.7 |
| **Number of dependents** | No dependents | 82 | 38.7 |
| | One or more | 130 | 61.3 |
| **Place of sleep** | Home (Owned or rented) | 102 | 48.1 |
| | Hotel/Street/Brothel | 30 | 14.2 |
| | Sleep at Friends and relatives | 79 | 37.3 |
| | Missing | 1 | 0.5 |

Table 2. Prevalence of alcohol use among female sex workers in Mbeya City, Tanzania.

| Characteristic | Categories | n | % |
|---|---|---|---|
| Ever used alcohol (N = 212) | Yes | 160 | 75.5 |
| | No | 52 | 24.5 |
| Alcohol use in the last 12 months[1] (N = 160) | No | 23 | 14.4 |
| | Yes | 137 | 85.6 |
| | Total | 160 | 100.0 |
| Alcohol use in the last 30 days[1] (N = 160) | No | 24 | 15.0 |
| | Yes | 136 | 85.0 |
| Alcohol Use Disorder (risky drinking) (N = 137) | No | 2 | 1.5 |
| | Yes | 135 | 98.5 |
| Heavy Episodic consumption (according TLFB Calendar) (N = 136) | No | 70 | 51.5 |
| | Yes | 66 | 48.5 |

Specifically, it was observed that a significant proportion of female sex workers, totalling 137 individuals (86.6%), reported engaging in alcohol use within the past year. Furthermore, an equally notable percentage of 136 individuals (85.2%) acknowledged recent alcohol consumption within the past month.

Remarkably, among those who reported alcohol use within the past year, a noteworthy number of cases, precisely 135 individuals (98.5%), displayed indicators of Alcohol Use Disorders (risky drinking). This highlights the considerable prevalence of problematic alcohol consumption patterns within this specific population. Furthermore, it is worth mentioning that a significant majority of female sex workers, accounting for 66%, reported engaging in binge drinking.

## Alcohol drinking patterns among female sex workers

When examining the alcohol drinking patterns among female sex workers, a fascinating picture emerges. A significant majority, encompassing 48.2% (n = 66) of the participants, engage in alcohol consumption at least four times per week. Moreover, an even more striking majority, constituting 57.7% (n = 79) of the individuals, reported consuming 10 or more drinks per drinking occasion.

A substantial portion of the participants, precisely 43.1% (n = 59), revealed to regularly consuming a minimum of six standard drinks per week. Furthermore, an intriguing revelation emerged, with 40.95% (n = 56) of the participants acknowledging a near-daily occurrence of consuming more than six standard drinks per occasion as reported in Table 3.

These findings bring to the forefront the captivating alcohol-drinking patterns prevalent among female sex workers. Such insights underscore the importance of comprehending the frequency and quantity of alcohol consumption within this specific population, illuminating the need for targeted interventions and support to address potential risks associated with these drinking patterns.

## Factors associated with alcohol use among female sex worker in the last 12 months

Table 4 reports on the presence of a relationship exhibited a protective effect on alcohol use (AOR = 0.19; 95% CI = 0.07–0.55), indicating that Female Sex Workers (FSWs) involved in relationships were less inclined to engage in alcohol consumption. On the other hand, individuals with incomes above the median demonstrated a higher likelihood of alcohol use (AOR = 1.19; 95% CI = 0.35–3.75), suggesting a potential association between income level and alcohol

**Table 3. Alcohol use patterns among female sex workers in Mbeya City.**

| Characteristic | Categories (N = 137) | n | % |
|---|---|---|---|
| **Alcohol use frequency** | Monthly or more | 3 | 2.2 |
| | 2–4 times a month | 18 | 13.1 |
| | 2–3 times a week | 50 | 36.5 |
| | 4 or more times a week | 66 | 48.2 |
| **Number of Drinks when drinking** | 1 or 2 | 3 | 2.2 |
| | 3 or 4 | 14 | 10.2 |
| | 5 or 6 | 17 | 12.4 |
| | 7 or 9 | 24 | 17.5 |
| | 10 or more | 79 | 57.7 |
| **Drinking six or More drinks per occasion** | Never | 1 | 0.7 |
| | Less than monthly | 4 | 2.9 |
| | Monthly | 17 | 12.4 |
| | Weekly | 59 | 43.1 |
| | Daily or almost daily | 56 | 40.9 |

consumption within the FSW population. Additionally, it was observed that FSWs with more than 10 sexual partners had a 1.62 times higher likelihood of alcohol use, underscoring a plausible link between alcohol consumption and the number of sexual partners in this particular group.

These findings offer valuable insights into the complex interplay of factors influencing alcohol use among female sex workers. The protective influence of being in a relationship, the potential influence of income level, and the association between sexual behaviour and alcohol use collectively contribute to understanding the dynamics surrounding alcohol consumption within this unique context.

## Reported alcohol use in the last 30 days among female sex workers and associated factors

Multivariate analysis was conducted to gain an understanding of the factors associated with reported alcohol use in the past month among female sex workers in Mbeya City, and

**Table 4. Reported alcohol use in the last 12 months among female sex workers and associated factors by multivariate analysis.**

| Characteristics | Categories | N | % | AOR | 95% CI |
|---|---|---|---|---|---|
| **Marital status** | Single | 113 | 91.1 | 1 | |
| | In a relationship (including married and cohabiting) | 24 | 66.7 | 0.19** | 0.07–0.55 |
| **Income** | Below median (90,000 and below) | 63 | 79.7 | 1 | |
| | Above median (>90000) | 74 | 91.4 | 1.19 | 0.38–3.75 |
| **Alcohol Use before sex** | Yes | 64 | 94.1 | 1 | |
| | No | 73 | 79.3 | 0.49 | 0.13–1.76 |
| **Number of sexual partners** | 1–10 partners | 65 | 81.3 | 1 | |
| | More than 10 partners | 72 | 90.0 | 1.62 | 0.56–4.67 |
| **Regretful sex after alcohol use** | Yes | 82 | 94.3 | 1 | |
| | No | 44 | 71.0 | 0.23* | 0.07–0.76 |
| **Observations** | | | | 149 | |

*** $p < 0.001$

** $p < 0.01$

* $p < 0.05$

**Table 5. Reported alcohol use in the last 30 days among female sex workers and associated factors.**

| Characteristic | Categories | N | % | AOR | 95% CI |
|---|---|---|---|---|---|
| **Level of education** | Incomplete primary school and never | 28 | 80 | 1 | |
| | Primary school | 46 | 80.7 | 1.05 | 0.27–4.06 |
| | Secondary school and above | 62 | 91.2 | 1.49 | 0.35–6.43 |
| **Marital status** | Single | 112 | 90.3 | 1 | |
| | In a relationship (including married and cohabiting) | 24 | 66.7 | 0.19** | 0.06–0.59 |
| **Income** | Median (90000 and below) | 61 | 77.2 | 1 | |
| | Above median income (>90000) | 75 | 92.6 | 1.33 | 0.36–4.90 |
| **dependents** | Yes | 94 | 88.7 | 1 | |
| | No | 42 | 77.8 | 0.57 | 0.19–1.68 |
| **Alcohol Use before sex** | Yes | 63 | 92.6 | 1 | |
| | No | 73 | 79.3 | 0.58 | 0.15–2.20 |
| **STI** | Yes | 27 | 96.4 | 1 | |
| | No | 109 | 82.6 | 0.35 | 0.03–3.98 |
| **Testing for HIV** | Yes | 115 | 83.3 | 1 | |
| | No | 21 | 95.5 | 1.38 | 0.14–13.98 |
| **Number of sexual partners** | 1–10 partners | 64 | 80 | 1 | |
| | More than 10 partners | 72 | 90 | 1.54 | 0.48–4.94 |
| **Regretful sex after alcohol use** | Yes | 84 | 96.6 | 1 | |
| | No | 42 | 67.7 | 0.13** | 0.03–0.57 |

*** p<0.001

** p<0.01, * p<0.05

the results are presented in **Table 5**. These findings provide valuable insights into the complex interplay of various factors contributing to alcohol consumption in this specific population.

The analysis revealed intriguing associations that shed light on the dynamics of alcohol use among female sex workers. Firstly, individuals with at least a primary education and incomes above the median were found to have a higher likelihood of consuming alcohol within the past 30 days. This suggests a possible relationship between educational attainment, income level, and alcohol consumption patterns in this context. Additionally, those who reported having more than 10 sexual partners showed an increased likelihood of alcohol use, although further exploration is necessary to fully comprehend the underlying factors driving this association (AOR = 1.54, 95% CI = 0.48–4.94).

On the contrary, being in a relationship emerged as a protection against alcohol use in the previous 30 days (AOR = 0.19, 95% CI = 0.06–0.59). This indicates that individuals involved in relationships were less inclined to engage in alcohol consumption during this timeframe. Furthermore, not having dependents was also associated with a lower likelihood of alcohol use (AOR = 0.57, 95% CI = 0.19–1.68), suggesting a potential protective influence against alcohol consumption.

These findings provide valuable insights into the nuanced factors influencing reported alcohol use among female sex workers in Mbeya City. The intricate relationship between education, income, relationship status, and dependents contributes to a deeper understanding of the complex dynamics that shape alcohol consumption behaviours in this specific population. By unravelling these associations, we can develop targeted interventions and support systems to address the challenges related to alcohol use in this context.

## Reported high episodic drinking among female sex worker and associated factors

In this section, we delve into the factors linked to (reported) high episodic drinking among female sex workers in Mbeya City, Tanzania. **Table 6** provides a comprehensive overview of these factors and their associations, shedding light on the unique dynamics within this context.

Our analysis reveals noteworthy findings regarding the variables influencing high episodic drinking among female sex workers in Mbeya City. Firstly, it suggests that those with a secondary school education or higher are 1.45 times more likely to engage in high episodic drinking. This highlights the potential role of education in shaping drinking patterns within this specific population.

Additionally, a strong correlation emerges between the non-use of condoms and heavy episodic drinking, showing that individuals who do not use condoms are 3.15 times more likely to be binge drinkers. This connection underscores the interplay between risky sexual behaviour and alcohol consumption among female sex workers.

Moreover, our analysis indicates that female sex workers with more than ten sexual partners have a 3.84 times higher likelihood of exhibiting high episodic drinking. Furthermore, those with an income above the median are also more prone to this behaviour. These findings emphasize the influence of both sexual behaviours and socioeconomic factors on high episodic drinking within the studied population.

**Table 6. Reported high episodic drinking among female sex workers and associated factors by multivariate analysis.**

| Characteristics | Categories | N | % | AOR | 95% CI |
|---|---|---|---|---|---|
| Level of education | Incomplete primary school and never | 19 | 65.5 | 1 | |
| | Primary school | 29 | 63 | 0.52 | 0.12–2.24 |
| | Secondary school and above | 55 | 88.7 | 1.45 | 0.32–6.47 |
| Marital status | Single | 91 | 80.5 | 1 | |
| | In a relationship (including married and cohabiting) | 12 | 50 | 0.41 | 0.12–1.38 |
| Income | Below median (90,000 and below) | 38 | 60.3 | 1 | |
| | Above median income | 65 | 87.8 | 1.70 | 0.50–5.80 |
| Dependents | Yes | 80 | 86 | 1 | |
| | No | 23 | 52.3 | 0.43 | 0.15–1.24 |
| Condom Use | Yes | 49 | 76.6 | 1 | |
| | No | 54 | 74 | 3.15 | 0.92–10.83 |
| STI | Yes | 17 | 65.4 | 1 | |
| | No | 86 | 77.5 | 0.60 | 0.10–3.47 |
| Testing for HIV | Yes | 96 | 82.1 | 1 | |
| | No | 7 | 35 | 0.23 | 0.05–1.08 |
| Number of sexual partners | 1–10 sexual partners | 36 | 55.4 | 1 | |
| | More than 10 partners | 67 | 93.1 | 3.84* | 1.13–13.03 |
| Regretful sex after alcohol use | Yes | 69 | 84.1 | 1 | |
| | No | 31 | 70.5 | 0.65 | 0.20–2.08 |
| Observations | | | | 126 | |

*** p<0.001

** p<0.01

* p<0.05

Conversely, our research uncovers a protective effect against high episodic drinking among female sex workers without dependents, as well as among those with dependents. This suggests that the presence of dependents may serve as a deterrent to engaging in heavy episodic drinking.

By unravelling these factors, our study contributes valuable insights into the complex dynamics of high episodic drinking among female sex workers in Mbeya City, Tanzania. This knowledge can guide the development of targeted interventions and strategies aimed at addressing the specific challenges associated with heavy episodic drinking in this particular population.

## Discussion

We sought to determine the prevalence of alcohol use and alcohol use disorders among female sex workers (FSWs) in Mbeya city, Tanzania. The female sex workers reported high alcohol consumption (86.6%) in the last 12 months. This prevalence is consistent with similar studies conducted in Dar es Salaam, Tanzania, Kenya and in other low and middle income countries [5, 23, 24]. Notably, among FSWs who reported alcohol use in the previous year, an alarming 98.5% were classified as having AUD, surpassing rates reported in previous studies conducted in Dar es Salaam, Tanzania and other Low and middle income countries [5, 24].

The study identified socio-demographic factors associated with alcohol use similar to those observed among young individuals in northern Tanzania. Notably, having an income higher than the median and being unmarried were significant predictors of alcohol use. Furthermore, the study revealed associations between alcohol use before engaging in sexual activity, regrettable sexual encounters following alcohol use, and current alcohol use. These findings corroborate research conducted in Mombasa, Kenya, and Ethiopia [8, 20].

Among the participants, the prevalence of alcohol use in the previous month stood at 85.0%, which aligns with findings reported in Kisumu, Kenya, Zambia, the US Virgin Islands and systematic reviews conducted in Mexican cities and other low and middle income countries [23, 25–28]. Notably, the prevalence of alcohol consumption within the last 30 days among Female Sex Workers in Mbeya City surpassed the rates reported among young female migrant workers in Kilimanjaro, northern Tanzania but it conforms with that reported in other low and middle income countries [23, 29].

The study uncovered several factors associated with alcohol use in the last 30 days. Income levels and educational attainment were influential, along with sexual behaviours such as having multiple sexual partners in the past month, engaging in alcohol use before engaging in sexual activity, and experiencing unfortunate sexual encounters following alcohol use. These findings align with studies conducted in Uganda, East Africa, and India, highlighting the correlation between alcohol consumption and engaging in risky behaviours [6, 26, 30, 31]. Moreover, this correlation emphasizes the indirect contribution of alcohol consumption to higher HIV prevalence among female sex workers. As a result, integrating alcohol use interventions into HIV/AIDS programs is imperative, as numerous studies conducted in East Africa have recommended.

The study findings revealed a significant prevalence of high alcohol consumption, with 48.5% of individuals who consume alcohol reporting high episodic drinking (HED). This proportion is considerably higher compared to rates observed in Kenya and Ethiopia [8, 25]. It is worth noting that Tanzania exhibits a higher per capita alcohol consumption rate (6.75) compared to Kenya (4.4) and Ethiopia (4.02) [8]. However, these results align with a study conducted in Uganda, where 53% of participants reported HED [32].

The study identified several factors associated with HED among female sex workers (FSW). Individuals with education beyond secondary school and an income above the median

demonstrated a significant association with HED. Conversely, FSWs without dependents and those in a committed relationship were less likely to engage in HED. These findings correlate with observations made in Kampala [6]. Additionally, individuals with HED were found to be three times more likely to engage in inconsistent condom usage, which is a concern for HIV prevention efforts. Similar findings regarding inconsistent condom usage have been reported in Ethiopia and Kampala [6, 8].

Responding to the World Health Organization's call to address harmful alcohol use, several policy interventions have been implemented in Sub-Saharan Africa [1, 33–35]. These interventions include measures like increased taxation on alcoholic beverages. However, more measures may be needed to adequately address the issue among vulnerable populations. A comprehensive approach involving multi-sectoral interventions targeting individual and socio-structural factors is necessary [5]. Incorporating alcohol intervention strategies into the healthcare system in Sub-Saharan Africa, including Tanzania, has been recommended [36, 37].

This study's findings should be interpreted in light of the following limitations: First, being a cross sectional study it was not possible to establish the factors causally associated with alcohol use and alcohol use disorder. Second, due to potential social desirability there was a potential underreporting of alcohol use and other risk factors that could bias the associations towards the null, however, that was unlikely for alcohol use because almost 90% of the female sex workers reported alcohol use in the last 12 months. Third, underreporting was also possible due to potential recall bias, however, this was minimized by limiting the reporting of alcohol use and other risk factors in the last 12 months and other tools, such as Timeline Follow Back calendar was limited to the last 30 days.

## Conclusion

The present study reveals a significant prevalence of alcohol use among female sex workers (FSW) in Mbeya, encompassing various aspects such as past-year use, heavy episodic use, and distinctive consumption patterns. Moreover, the prevalence of alcohol use disorder within this population is notable. It is evident that targeted interventions to reduce alcohol consumption are imperative for FSWs who reported alcohol use in the previous year. Notably, factors such as higher income and being single (unmarried) are associated with alcohol consumption, while engaging in risky sexual behaviours such as regrettable sexual activity, multiple sexual partners, and consuming alcohol prior to sexual encounters are correlated with alcohol use.

## Supporting information

**S1 Checklist. PLOS' questionnaire on inclusivity in global research.**
(DOCX)

## Acknowledgments

I would like to express my deep gratitude to all study participants for their time and willingness to participate which made this study possible.

I would also like to thank Dr. Wiston William and Ms Doreen Pamba, for their advices and guidance during implementation of this study and supervising data collection.

I would also like to extend my thanks to the research assistants (Ms Zeituni Mchomvu and Bareke Msomba) at NIMR Mbeya for their assistance during data collection and my employer Amref health Africa for allowing me to conduct this study.

## Author Contributions

**Conceptualization:** Andrew Kapaya Augustine, Lucas Maganga, Joel Msafiri Francis.

**Data curation:** Andrew Kapaya Augustine, Joel Msafiri Francis.

**Formal analysis:** Andrew Kapaya Augustine, Joel Msafiri Francis.

**Investigation:** Andrew Kapaya Augustine, Lucas Maganga, Joel Msafiri Francis.

**Methodology:** Andrew Kapaya Augustine, Lucas Maganga, Joel Msafiri Francis.

**Project administration:** Andrew Kapaya Augustine, Lucas Maganga.

**Resources:** Lucas Maganga.

**Software:** Joel Msafiri Francis.

**Supervision:** Lucas Maganga, Joel Msafiri Francis.

**Validation:** Andrew Kapaya Augustine, Joel Msafiri Francis.

**Writing – original draft:** Andrew Kapaya Augustine.

**Writing – review & editing:** Andrew Kapaya Augustine, Lucas Maganga, Joel Msafiri Francis.

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
