## [Decision Letter · Decision Letter 0]

20 Feb 2024

PGPH-D-23-02419

Epidemiology of alcohol use and alcohol use disorder among female sex workers in Mbeya City, Tanzania.

Dear Dr.  Andrew Kapaya Augustine,

Thank you for submitting your manuscript to PLOS Global Public Health. After careful consideration, we feel that it has merit but does not fully meet PLOS Global Public Health’s publication criteria as it currently stands. Therefore, we invite you to submit a revised version of the manuscript that addresses the points raised during the review process.

We look forward to receiving your revised manuscript.

Kind regards,

Md Nazmul Huda, BSS, MSS, PhD

Academic Editor

Journal Requirements:

2. Tables should not be uploaded as individual files. Please remove these files and include the Tables in your manuscript file as editable, cell-based objects. For more information about how to format tables, see our guidelines:

https://journals.plos.org/globalpublichealth/s/tables

Additional Editor Comments (if provided):

Reviewers' comments:

Reviewer's Responses to Questions

**Comments to the Author**

1. Does this manuscript meet PLOS Global Public Health’s publication criteria? Is the manuscript technically sound, and do the data support the conclusions? The manuscript must describe methodologically and ethically rigorous research with conclusions that are appropriately drawn based on the data presented.

Reviewer #1: Yes

Reviewer #2: Partly

Reviewer #3: Partly

2. Has the statistical analysis been performed appropriately and rigorously?

Reviewer #1: Yes

Reviewer #2: Yes

Reviewer #3: No

3. Have the authors made all data underlying the findings in their manuscript fully available (please refer to the Data Availability Statement at the start of the manuscript PDF file)?

Reviewer #1: Yes

Reviewer #2: Yes

Reviewer #3: No

4. Is the manuscript presented in an intelligible fashion and written in standard English?

Reviewer #1: Yes

Reviewer #2: No

Reviewer #3: No

5. Review Comments to the Author

Reviewer #1: The present study offers valuable insights into the targeted population, which could aid the government in developing more effective prevention plans. However, it is imperative to take into account the limitations and issues that have been identified and attached. These constraints are significant and merit careful consideration.

Reviewer #2: Dear EIC,

Thank you for your kind invitation,

The manuscript was conducted to evaluate the prevalence, and factors associated with alcohol use among sex workers in Mbeya, Tanzania. the subject was interesting. The studied sample is part of the other study in a cohort, who have a high risk for AIDS. The probability of selection bias is highly important. To estimating the sample size, what is the prevalence of alcohol consumption in the target population? Please specify with related reference.

The most important problem in this study is cross sectional design and high probability of recall bias, specially in these high risk women.

Reviewer #3: -The paragraph of the introduction is long and most sections of the manuscript are too complicated.

-Some reference numbers are in (), and some in [].

-The order of reference numbers in the text is not observed. for example: For example, reference number 11 is inserted after references 3 , 4 and before reference 5.

- In Sample Size estimation section, The authors should have mentioned the values of p and q and their references.

- The authors used abbreviation of female sex workers in *Factors associated with alcohol use* section, While it should have been used earlier in the text.

-What were the criteria for classifying people in terms of income?

- The representativeness of the sample is not well defined.

-Several analyzes were performed to investigate the relationship,While the mentioned sample size formula belongs to descriptive studies. In addition to this, there are issues related to multiple comparisons.

- Providing a clear definition of Alcohol Use Disorder and other variables is essential.

-The number of references used in the text does not match the number of references at the end of the article.

- To avoid problems related to multiple comparisons, use appropriate models that allow simultaneous examination of the effect of independent variables on several response variables.

6. PLOS authors have the option to publish the peer review history of their article (what does this mean?). If published, this will include your full peer review and any attached files.

**Do you want your identity to be public for this peer review?** For information about this choice, including consent withdrawal, please see our Privacy Policy.

Reviewer #1: **Yes: **Farima Safari

Reviewer #2: No

Reviewer #3: No

---

## [Editor Report · Decision Letter 1]

10 Apr 2024

Epidemiology of alcohol use and alcohol use disorder among female sex workers in Mbeya City, Tanzania.

PGPH-D-23-02419R1

Dear Dr Andrew Kapaya Augustine,

We are pleased to inform you that your manuscript 'Epidemiology of alcohol use and alcohol use disorder among female sex workers in Mbeya City, Tanzania.' has been provisionally accepted for publication in PLOS Global Public Health.

Best regards,

Md Nazmul Huda, PhD

Academic Editor